# Differential Expression of Small Non-Coding RNAs in Uterine Leiomyomas

**DOI:** 10.3390/ijms26041688

**Published:** 2025-02-16

**Authors:** Tsai-Der Chuang, Nhu Ton, Shawn Rysling, Daniel Baghdasarian, Omid Khorram

**Affiliations:** 1The Lundquist Institute for Biomedical Innovation, Torrance, CA 90502, USA; tchuang@lundquist.org (T.-D.C.); nhu.ton@lundquist.org (N.T.); shawn.rysling@lundquist.org (S.R.); 2Department of Obstetrics and Gynecology, Harbor-UCLA Medical Center, Torrance, CA 90502, USA; dbaghdasarian@dhs.lacounty.gov; 3Department of Obstetrics and Gynecology, David Geffen School of Medicine at University of California, Los Angeles, CA 90095, USA

**Keywords:** leiomyoma, fibroid, sncRNA, race, MED12, piRNA, miRNA

## Abstract

We performed next-generation sequencing (NGS) on RNA from 19 paired leiomyoma (Lyo) and myometrium (Myo) specimens, stratified by race/ethnicity (White: n = 7; Black: n = 12) and mediator complex subunit 12 (MED12) mutation status (mutated: n = 10; non-mutated: n = 9). Analysis identified 2,189 small non-coding RNAs (sncRNAs) with altered expression in Lyo compared to paired Myo (≥1.5-fold change), including small nuclear RNAs (snRNAs), small nucleolar RNAs (snoRNAs), microRNAs (miRNAs), and PIWI-interacting RNAs (piRNAs). Among these, 17 sncRNAs showed differential expression in the MED12-mutated group versus Myo, while minimal changes were observed in the non-mutated group. Additionally, 31 sncRNAs displayed differential expression in Black women compared to White women. For validation, five novel miRNAs (miR-19a-3p, miR-99a-5p, miR-3196, miR-499a-5p, and miR-30d-3p) and five piRNAs (piR-009295, piR-020326, piR-020365, piR-006426, and piR-020485) were analyzed in 51 paired Lyo samples using qRT-PCR. Reduced expression of the selected sncRNAs was confirmed in Lyo versus Myo, with miR-19a-3p, miR-3196, miR-30d-3p, piR-006426, and piR-020485 linked to MED12 status, while miR-499a-5p and miR-30d-3p were associated with race/ethnicity. These findings suggest that sncRNA dysregulation contributes to altered gene expression in Lyo, influenced by MED12 mutation and racial background.

## 1. Introduction

Leiomyomas (Lyo) are benign fibrotic tumors of the uterus affecting approximately 70% of women during their reproductive years. The growth of Lyo is dependent on ovarian steroids. These tumors are a significant health burden and, due to the limited availability of effective medical treatments, account for more than 200,000 hysterectomies performed annually in the United States [1,2]. Symptomatic leiomyomas are a major cause of chronic pelvic pain, abnormal uterine bleeding, and infertility, with a higher prevalence and symptom severity observed in Black women [3,4]. Genome-wide association studies have identified numerous genes associated with leiomyoma size and volume in a race-dependent manner [5,6].

Lyo are characterized by the excessive accumulation of extracellular matrix (ECM) and altered expression of various coding and non-coding RNAs (ncRNAs), including long non-coding RNAs (lncRNAs) and small non-coding RNAs (sncRNAs) [7,8,9,10,11]. Genetic heterogeneity, including chromosomal rearrangements and driver mutations in several genes, including MED12, has been implicated in Lyo development and progression [2]. MED12 mutation in exon 2 is the most common mutation and is present in up to 80% of Lyo. This mutation has been linked to the abnormal activation of Wnt/β-catenin signaling, sex steroid receptor signaling, and the dysregulation of cell cycle- and fibrosis-related gene expression [12,13].

Emerging evidence indicates that sncRNAs, particularly microRNAs (miRNAs), play crucial roles in regulating protein-coding gene expression via transcriptional, post-transcriptional, and epigenetic mechanisms [14]. SncRNAs are transcribed from precursor molecules ranging from 60 to 200 nucleotides and, upon maturation, are classified into distinct groups: microRNAs (miRNAs; 17–22 nucleotides), small nuclear and small nucleolar RNAs (snRNAs and snoRNAs; 70–120 nucleotides), and PIWI-interacting RNAs (piRNAs; 26–33 nucleotides). SnoRNAs are highly conserved and play a crucial role in RNA modification and processing. Based on their structural features, they are further categorized into two classes: C/D box snoR-NAs (SNORDs) and H/ACA box snoRNAs (SNORAs) [15,16]. PiRNAs are the largest sncRNA class and regulate gene expression epigenetically and post-transcriptionally [15]. snRNAs and snoRNAs can be processed into miRNAs or siRNAs, forming RNA-induced silencing complexes (RISCs) for mRNA cleavage, degradation, or translational inhibition [11,16].

sncRNAs, in a cell- and tissue-specific manner, regulate various physiological processes, and their dysregulation has been linked to numerous disorders, including tumorigenesis and tissue fibrosis [17,18,19]. The aberrant expression of several miRNAs in Lyo that are functionally associated with angiogenesis, inflammation, ECM accumulation, and fibrosis, such as let-7, miR-21, miR-29, miR-200c, miR-93, and miR-135b, has previously been reported [20,21,22,23,24,25,26,27,28,29]. However, the roles and expression profiles of other sncRNA members in Lyo remain largely unexplored.

The impact of the differential expression of sncRNAs in Lyo pathogenesis remains unknown. Therefore, this study aimed to elucidate the expression profiles of differentially expressed sncRNAs in a large cohort of Lyo and matched myometrium samples using high-throughput sequencing. The analysis included stratification by MED12 mutation status and race/ethnicity. Validation studies of 10 differentially expressed sncRNAs were conducted in 51 Lyo-matched pairs using qRT-PCR. We hypothesized that a unique set of sncRNAs, differentially expressed in Lyo, correlates with MED12 mutation status and race/ethnicity, providing potential mechanistic insights into Lyo pathogenesis.

## 2. Results

### 2.1. High-Throughput Sequencing of sncRNA Expression in Paired Leiomyomas and Myometrium

Using high-throughput next-generation RNA sequencing (NGS), we previously identified differential expression profiles of MED12 mutation- and race/ethnicity-associated lncRNAs and coding transcripts in Lyo and paired Myo [7,8,9]. In this study, we performed NGS on RNA extracted from 19 paired Lyo and Myo specimens, stratified by race/ethnicity (White: n = 7; Black: n = 12) and MED12 mutation status (mutated: n = 10; non-mutated: n = 9). After normalization, hierarchical clustering, and TreeView analysis, we identified 2189 sncRNAs (14 snRNAs, 91 snoRNAs, 875 miRNAs, 478 piRNAs, 30 RNA pseudogenes, and 701 unclassified) with altered expression in Lyo compared to paired Myo (≥1.5 fold change, either up or down; Figure 1A). Volcano plot analysis (fold change ≥1.5, *p* < 0.05; Figure 1B) and principal component analysis (PCA) demonstrated distinct RNA transcript patterns between Lyo and Myo, and k-means clustering confirmed the reliability of the data (Figure 1C). Gene Ontology (GO) and KEGG pathway enrichment analyses indicated that these altered sncRNAs were predominantly involved in pathways such as proteoglycans in cancer and MAPK signaling (Figure 1D).

We next examined the impact of MED12 mutation status on sncRNA expression. Fold-change analysis (Lyo/Myo) revealed 343 (1 snRNA, 18 snoRNAs, 215 miRNAs, 78 piRNAs, 3 RNA pseudogenes, and 28 unclassified) sncRNAs with altered expression in the MED12-mutated group (≥1.5-fold change, either up or down). Hierarchical clustering and TreeView analysis categorized these transcripts into distinct groups (Figure 2A). Seventeen sncRNAs displayed differential expression uniquely in the MED12-mutated group, with minimal changes in the non-mutated group, as shown in the heatmap (Figure 2B). GO and KEGG analyses of these 17 sncRNAs highlighted their roles in signal transduction by growth factor receptors and the PI3K-Akt signaling pathway (Figure 2C). Race/ethnicity-based comparisons identified 293 (11 snoRNAs, 160 miRNAs, 89 piRNAs, 5 RNA pseudogenes, and 28 unclassified) sncRNAs with altered expression (≥1.5-fold change, either up or down) in Black women compared with White women. Hierarchical clustering and TreeView analysis categorized these sncRNAs into distinct groups (Figure 3A). Thirty-one sncRNAs were uniquely differentially expressed in Black women, with no significant changes in White women (Figure 3B). GO and KEGG pathway analyses revealed that these sncRNAs were enriched in pathways related to DNA-binding transcription activator activity and interleukin signaling (Figure 3C).

### 2.2. Validation of sncRNA Expression by qRT-PCR

To validate the RNA sequencing results, 10 sncRNAs previously studied in other disease models were selected for validation in 51 paired Lyo and Myo tissues from premenopausal patients not receiving hormonal medications. Since most differentially expressed sncRNAs identified in this study were miRNAs and piRNAs, we did not select any snRNAs or snoRNAs for validation. Based on RNA sequencing analysis comparing Lyo and Myo tissues (≥1.5-fold change, either up or down), miR-19a-3p, miR-99a-5p, piR-009295, piR-020326, and piR-020365 were downregulated, while piR-006426 and piR-020485 were upregulated. These sncRNAs were chosen for validation by qRT-PCR, which confirmed the significant downregulation of all selected miRNAs and piRNAs in Lyo compared to paired Myo (*p* < 0.05; Figure 4). However, the expression patterns of piR-006426 and piR-020485 observed in qRT-PCR did not fully align with the RNA sequencing results. In the context of MED12 mutation status, RNA sequencing identified significant alterations in miR-19a-3p, miR-99a-5p, miR-3196, miR-499a-5p, miR-30d-3p, piR-020485 (all downregulated), and piR-006426 (upregulated) when comparing mutant (Lyo/Myo) to wild-type (Lyo/Myo) tissues (≥1.5-fold change). Validation by qRT-PCR showed that miR-99a-5p, miR-3196, miR-30d-3p, piR-006426, and piR-020485 were significantly downregulated in the MED12 mutant group (Figure 5). Further analysis stratified by race/ethnicity revealed the significant differential expression of miR-499a-5p and miR-30d-3p in Lyo tissues, with higher expression levels observed in Hispanic women compared to Black women (Figure 6). However, qRT-PCR expression patterns did not fully match RNA sequencing results. Specifically, RNA sequencing identified significant alterations in miR-30d-3p and piR-020485 (upregulated), piR-009295, piR-020326, piR-020365, and piR-006426 (all downregulated) when comparing Black and White groups (Lyo/Myo, ≥1.5-fold change).

## 3. Discussion

Through high-throughput RNA sequencing, we previously reported the expression profiles of numerous lncRNAs alongside protein-coding RNAs in Lyo and paired Myo [7,8,9,10]. In the present study, using RNA sequencing data annotated with the AASRA pipeline, miRBase, piRBase, the Genomic tRNA database, and ENSEMBL, we identified novel sncRNA transcripts that were differentially expressed in Lyo. Of these, a subset of sncRNAs were significantly altered in MED12-mutated Lyo and in Lyo from Black women, with minimal changes in non-mutated Lyo and Lyo from White women. From this dataset, we selected 10 sncRNAs (miR-19a-3p, miR-99a-5p, miR-3196, miR-499a-5p, miR-30d-3p, piR-009295, piR-020326, piR-020365, piR-006426, and piR-020485) for validation using qRT-PCR in 51 paired Lyo specimens. The qRT-PCR analysis partially confirmed the RNA sequencing findings and indicated that the expression of sncRNAs was influenced by MED12 mutation status (miR-19a-3p, miR-3196, miR-30d-3p, piR-006426, and piR-020485) and race/ethnicity (miR-499a-5p and miR-30d-3p).

Our study revealed the partial confirmation of RNA sequencing results by qRT-PCR, with certain sncRNAs displaying discrepancies between the two methodologies. Several potential factors could explain these differences. Firstly, RNA-seq experiments can have high variance, especially for genes expressed in low amounts, and this may affect the accuracy of expression estimates and subsequent validation efforts [30]. This variation can stem from differences in sample preparation, library preparation batch effects, alignment protocols, and sequencing technology biases, which may introduce biases or false positives, particularly for sncRNAs with complex secondary structures [30]. In contrast, qRT-PCR requires specific primer design, which can be challenging for sncRNAs, potentially leading to suboptimal amplification efficiency and variability in expression levels [31]. Additionally, discrepancies may also arise due to differences in normalization strategies. RNA-seq data are typically normalized using bioinformatics tools that account for sequencing depth and library size, whereas qRT-PCR relies on the selection of appropriate endogenous controls, which may vary across samples and conditions [31]. Overall, these findings suggest that while RNA-seq is a powerful tool for transcriptome analysis, due to the high variance, additional validation studies are necessary to ensure accuracy.

Consistent with previous publications [10,20,32,33,34], our sncRNA sequencing analysis of miRNA profiling revealed a similar trend in the differential expression of miRNAs such as miR-21, miR-34a, the Let-7 family, the miR-200 family, miR-23b, and the miR-29 family [10,20,32,33,34], highlighting their potential key roles in Lyo pathogenesis. Although information on the regulatory mechanisms behind the sncRNAs identified in this study in Lyo is limited, several have been implicated in tumorigenesis and other pathological processes. MiR-19a, a member of the miR-17-92 cluster, plays critical roles in various biological pathways and diseases. It enhances JAK-STAT signaling by targeting Suppressor of Cytokine Signaling 3 (SOCS3), thereby influencing inflammatory responses [35]. MiR-19a also promotes cell proliferation and migration by targeting the tumor suppressor T-cell intracellular antigen 1 (TIA1) in colorectal cancer [36] and is upregulated in bladder cancer, where it correlates with aggressive tumor phenotypes [37]. Beyond oncology, miR-19a-3p has demonstrated potential as a therapeutic agent in cardiovascular disease, inhibiting endothelial dysfunction in atherosclerosis by modulating the Hippo/YAP signaling pathway through Junctional Protein Associated with Coronary Artery Disease (JCAD) [38]. Additionally, miR-19a/19b enhances cardiac repair and protects against myocardial infarction, underscoring its broader therapeutic potential [39]. MiR-99a has been widely studied for its regulatory roles in cancer progression, cardiovascular health, and cellular functions. In breast cancer, miR-99a inhibits tumor growth by targeting the mammalian target of rapamycin (mTOR) pathway, reducing cell viability and promoting apoptosis [40]. It also suppresses cancer progression by targeting fibroblast growth factor receptor 3 (FGFR3), thereby reducing cell proliferation and invasion [41]. In endothelial cells, miR-99a modulates inflammation via the regulation of the NF-κB pathway, further emphasizing its role in cardiovascular health [42].

MiR-3196 acts as a tumor suppressor in gastric cancer, where its downregulation is associated with lymph node metastasis and advanced tumor node metastasis (TNM) stages [43]. The lncRNA MAFG-AS1 has been shown to promote cell proliferation and migration in drug-resistant hepatocellular carcinoma (HCC) and pancreatic cancer by sponging miR-3196, leading to the upregulation of STRN4 and NFIX, respectively [44,45]. MiR-499a has been extensively studied in cancer and cardiovascular contexts. In prostate cancer, its overexpression inhibits proliferation and induces apoptosis by targeting UBE2V2, demonstrating tumor-suppressive properties [46]. In acute myeloid leukemia, miR-499a-5p suppresses the progression of the disease by promoting the ubiquitin-proteasomal degradation of METTL3, thereby reducing m6A modification levels [47]. Its neuroprotective role has been highlighted in studies showing that miR-499a-5p attenuates cerebral ischemia/reperfusion injury by targeting PDCD4 [48]. Furthermore, miR-499a-5p is a potential biomarker for acute myocardial infarction and risk stratification in endometrial cancer, where its expression correlates with patient prognosis [49,50]. MiR-30d, a member of the miR-30 family, has diverse roles in cancer biology. In ovarian cancer, it acts as an oncomir, promoting tumor cell proliferation and migration, with its chromosomal locus amplified in over 30% of solid tumors, and is associated with poor clinical outcomes [51]. Conversely, in pancreatic cancer, miR-30d suppresses the proliferation and invasiveness of the cancer by targeting the SOX4/PI3K-AKT axis, with its downregulation linked to poor patient prognosis [52]. Additionally, the lncRNA circPVT1 promotes lung squamous cell carcinoma proliferation by sponging miR-30d [53]. Collectively, these findings emphasize the diverse roles of sncRNAs in tumorigenesis and other diseases, underscoring their potential as therapeutic targets and diagnostic biomarkers.

PiRNAs are derived from long RNA precursors through a dicer-independent mechanism and are transcribed from the 3′ untranslated regions of intergenic non-protein-coding and protein-coding genes [54]. Initially identified in gonadal tissues, piRNAs are now known to be expressed in somatic tissues, circulating cells, and cancer stem cells [55,56]. Functionally, piRNAs interact with PIWI proteins to regulate gene expression through translational control, transposon silencing, and epigenetic mechanisms, such as telomere regulation, via specific methyltransferases or histone modifications [57,58,59,60]. The aberrant expression of PIWI proteins has been reported in various cancers [61,62,63]. In Lyo, AGO2 expression is significantly elevated, suggesting increased levels of post-transcriptional RNA processing [11]. Specific piRNAs, such as piR-009295, have been linked to cancer progression. For example, piR-009295 is associated with epithelial–mesenchymal transition (EMT) processes in ovarian cancer, suggesting a role in tumor progression [64]. Additionally, piR-009295 has been identified as a differentially expressed piRNA in colorectal cancer (CRC), with studies exploring its potential as a diagnostic biomarker using machine learning-based tools [65]. Other piRNAs, such as piR-020326, have been evaluated for their diagnostic utility in endometriosis, with varying levels of diagnostic performance reported [66]. Similarly, piR-020365 and a panel of four other piRNAs (piR-001311, piR-004153, piR-017723, and piR-017724) have shown promise as non-invasive biomarkers for CRC detection [67]. The elevated expression of piR-020365 has also been observed in breast tumors and lymph node metastases, further emphasizing its role in cancer biology [68]. The piRNA piR-006426 has been implicated in cardiovascular disease. It is significantly downregulated in the serum exosomes of heart failure patients, suggesting a role in disease pathogenesis and its potential as a biomarker [69]. In reproductive biology, piR-020485 has been linked to embryo quality and viability. Studies have shown that piR-020485 is associated with developmental potential at the morula stage and implantation success at the blastocyst stage, making it a promising biomarker for assisted reproductive technologies [70,71]. Additionally, piR-020485 is part of the core RNA repertoire in human serum, suggesting roles in intercellular communication and broader physiological or pathological contexts [72]. The differentially expressed piRNAs (piR-009295, piR-020326, piR-020365, and piR-006426) were also identified in our previous sncRNA profiling study using a small sample set (n = 3) and subsequently validated in a larger cohort of paired specimens (n = 20) [11].

Our GO and KEGG pathway enrichment analyses revealed that MAPK signaling and proteoglycan-associated pathways are predominantly involved in the differential expression of sncRNAs in Lyo. Proteoglycans are key components in the excessive ECM accumulation in Lyo [3]. Among the proteoglycans, versican is abundantly expressed in Lyo [73,74]. The differences in proteoglycan composition may contribute to the pathological expansion of fibrotic tissue through mechanisms involving excessive matrix deposition, matrix disorganization, cell proliferation, and dysregulated TGF-β signaling [74]. Additionally, the aberrant activation of MAPK signaling is frequently observed in Lyo, leading to the increased expression of growth factors, extracellular matrix proteins, and other molecules that contribute to the development and maintenance of Lyo [75]. Estrogen, a primary hormone associated with Lyo growth, can activate the MAPK pathway through its receptor (ER) [75]. Furthermore, MAPK signaling exhibits cross-talk with other pathways, such as the PI3K/Akt pathway, further complicating the regulatory mechanisms that govern Lyo cell behavior [76]. Additionally, the signal transduction by growth factor receptors and the PI3K-Akt signaling pathways was identified as being associated with MED12 mutation-specific sncRNAs in Lyo by GO and KEGG pathway enrichment analyses. Growth factors such as platelet-derived growth factor (PDGF) and epidermal growth factor (EGF) bind to receptors on Lyo cells, triggering a signaling cascade that activates phosphoinositide 3-kinase (PI3K) and produces phosphatidylinositol-3,4,5-trisphosphate (PIP3). This recruits and activates Akt, or Protein Kinase B, a key downstream effector [75]. Phosphorylated Akt promotes cell proliferation by enhancing cell cycle progression and inhibiting apoptosis. It also activates the mTOR pathway to stimulate protein synthesis and regulates collagen production, contributing to extracellular matrix buildup [75]. Collectively, these processes contribute to the growth and development of Lyo.

In addition, GO and KEGG pathway analyses also revealed that racially specific sncRNAs were enriched in pathways related to DNA-binding transcription activator activity and interleukin signaling. Several studies have highlighted DNA-binding transcription activators and associated pathways in Lyo development. Activator protein-1 (AP-1) is enriched in differentially acetylated enhancers, and AP-1 motifs are found in promoters of dysregulated genes, indicating direct AP-1 target involvement [77]. HMGA1 and HMGA2 are overexpressed in 20-30% of fibroids and linked to altered DNA methylation patterns. MED12-mutant and HMGA2-high Lyo show distinct CpG methylation changes, including HMGA1 promoter hypermethylation and HMGA2 gene body hypomethylation [78]. Transcription factors like NR2F2 and CTNNB1 (β-catenin) are aberrantly expressed, with β-catenin acting as a key Wnt pathway co-activator [79]. Interleukin signaling pathways play a critical role in the development and progression of Lyo. Interleukin-33 (IL-33) has a profibrotic effect, with serum IL-33 levels positively correlating with Lyo weight and size, suggesting its involvement in Lyo growth [80]. Other interleukins, including IL-1, IL-11, IL-13, and IL-15, contribute to Lyo growth through interactions with estrogen and progesterone [81]. Bioinformatics analysis of differentially expressed genes (DEGs) in fibroids further indicates that many DEGs are enriched in cytokine-related pathways, emphasizing the importance of interleukin signaling in fibroid progression and identifying potential therapeutic targets [7,8].

Recent studies have highlighted the role of MED12 in tumor initiation, as demonstrated by the development of Lyo-like tumors in transgenic mice with uterine-specific MED12 mutations [12,82]. The Mediator complex, comprising the head, middle, tail, and kinase modules, regulates gene expression by transducing signals from gene-specific transcription factors to RNA polymerase II [83]. A potential mechanism behind the widespread effects of MED12 mutations involves its interaction with super-enhancers that regulate entire gene families [77]. Our previous work demonstrated that super-enhancer expression in Lyo is frequently altered and depends on MED12 mutation status [84]. MED12 mutations drive transcriptional dysregulation by altering enhancer architecture and promoting R-loop formation, linking transcriptional regulation to genomic instability [85]. MED12-mutated Lyo are typically smaller and more likely to be subserosal [86]. Conversely, MED12 mutation-negative Lyo exhibit copy number alterations in other Mediator complex subunits, such as CDK8, MED18, MED8, and the lncRNA RNA340 (CASC15). Mutations in MED12 also disrupt its ability to activate cyclin C-dependent CDK8 and CDK19, contributing to dysregulated signaling [87,88]. Furthermore, silencing MED12 in human uterine fibroid cell lines inhibits Wnt/β-catenin signaling, sex steroid receptor signaling, and pathways associated with cell cycle progression and fibrosis [89]. In line with the data presented here, our previous report demonstrated a significant downregulation of the miR-29 family in Lyo in an MED12-mutation-dependent manner [90]. Race also plays a critical role in the progression and severity of Lyo, with Black women experiencing larger, more numerous fibroids with earlier onset and with greater symptom severity compared to other racial groups [5,91]. Several biological mechanisms have been proposed to explain these disparities, including a higher prevalence of vitamin D deficiency, increased aromatase (CYP19) and progesterone receptor-A (PR-A) levels, and reduced retinoic acid receptor-α (RARA) expression in Black women [92,93]. Additionally, genetic polymorphisms in estrogen synthesis-related genes, such as ER, CYP17, and COMT, are more common in Lyo in Black women [92,93], potentially contributing to differential disease progression. Emerging evidence also highlights the race-specific dysregulation of sncRNAs in Lyo pathophysiology. The differential expression of miRNAs, including miR-200c, miR-21, and Let-7, has been reported in Lyo from Black women [20,33]. Additionally, variations in DNA methylation profiles between myometrial samples from Black and White women have been identified [94], which can influence the expression of miRNAs and other sncRNAs involved in fibroid pathogenesis. Furthermore, given that fibroids are hormone-responsive, racial differences in hormone-related gene expression further exacerbate these disparities [95]. A deeper understanding of these molecular mechanisms is crucial in developing targeted interventions and personalized treatment strategies to address racial disparities in Lyo pathophysiology.

In conclusion, our findings provide a comprehensive profile of differentially expressed sncRNAs in Lyo and point to the significant impact of race/ethnicity and MED12 mutation status as variables that further influence their expression. The aberrant expression of sncRNAs, likely through dynamic regulatory interactions with other non-coding RNAs (e.g., lncRNAs) and protein-coding genes, may influence key cellular processes such as cell proliferation, ECM accumulation, and inflammation, contributing to the development and progression of Lyo. Future research is needed to further elucidate the expression patterns, regulatory networks, and molecular mechanisms underlying the roles of sncRNAs in Lyo pathobiology.

## 4. Materials and Methods

### 4.1. Myometrium and Leiomyoma Tissue Collection

Portions of uterine leiomyomas (intramural, 3–5 cm in diameter) and paired myometrium (n = 51) were collected from patients at Harbor-UCLA Medical Center. The study was approved by the Institutional Review Board of the Lundquist Institute (18CR-31752-01R), and informed consent was obtained from all participants. Patients had not taken any hormonal medications for at least three months prior to surgery. The paired tissue samples were obtained from women aged 30–53 years (mean age, 44 ± 5.5 years) and included individuals of Caucasian (n = 6), African American (n = 12), Hispanic (n = 29), and Asian (n = 4) descent. The tissues were snap-frozen and stored in liquid nitrogen for subsequent analysis, as previously described [7,8,9]. The patient demographics are listed in Appendix A.

### 4.2. MED12 Mutation Analysis

Genomic DNA was extracted from 100 mg of freshly frozen Lyo and paired myo tis-sues using the MagaZorb DNA Mini-Prep Kit (Promega, Madison, WI, USA) following the manufacturer’s instructions. PCR amplification and Sanger sequencing (Laragen Inc., Culver City, CA, USA) were performed to analyze MED12 exon 2 mutations. The primers used for amplification were as follows: sense, GCCCTTTCACCTTGTTCCTT; antisense, TGTCCCTATAAGTCTTCCCAACC. PCR products were sequenced using Big Dye Terminator v3.1 chemistry, and the sequences were analyzed with ChromasPro 2.1.8 software and compared to the MED12 reference sequences (NG_012808 and NM_005120). Among the 19 paired tissues used for next-generation RNA sequencing, 10 Lyo were MED12 mutation-positive, and 9 were MED12 mutation-negative. Analysis of all specimens (n = 51) revealed that 33 Lyo (64.7%) harbored MED12 mutations, while no mutations were detected in the paired myometrial samples. Missense mutations in MED12 exon 2 were the most common alteration (30/33 pairs), followed by in-frame insertion–deletion mutations (3/33 pairs). The specific missense mutations identified in exon 2 included c.130G>C (p.Gly44Arg) (2/30 pairs), c.130G>A (p.Gly44Ser) (6/30 pairs), c.130G>T (p.Gly44Cys) (2/30 pairs), c.131G>C (p.Gly44Ala) (3/30 pairs), c.131G>A (p.Gly44Asp) (9/30 pairs), c.131G>T (p.Gly44Val) (7/30 pairs), and c.128A>C (p.Gln43Pro) (1/30 pairs).

### 4.3. RNA Sequencing and Bioinformatics Analysis

Total RNA was extracted from leiomyomas and matched myometrium using TRIzol (Thermo Fisher Scientific Inc., Waltham, MA, USA). RNA concentration and integrity were assessed with a Nanodrop 2000c spectrophotometer (Thermo Scientific, Wilmington, DE, USA) and an Agilent 2100 Bioanalyzer (Agilent Technologies, Santa Clara, CA, USA), as described previously [7,8,9]. Only samples with RNA integrity numbers (RINs) ≥ 9 were selected for library preparation. For small RNA sequencing library preparation, 500 ng of total RNA from each tissue was used following the protocol provided with the TruSeq Small RNA Kit (Illumina, San Diego, CA, USA). Libraries were pooled and sequenced on an Illumina MiSeq platform with a single-ended 35 bp run, achieving approximately 10 million reads per library and an alignment rate of 80–90% [11]. Data visualization included hierarchical clustering and TreeView graphs, volcano plots, PCA plots, and pathway enrichment analysis plots, generated using Flaski [96] and RNAenrich [97]. Differential gene expression analysis met all statistical quality thresholds and was deemed suitable for subsequent analyses. We used 1.5-fold change instead of 2.0. to identify a greater number of differentially expressed genes. We analyzed the data with robust statistical tests including adjusted p-values and FDR corrections. This approach can help balance the trade-off between identifying too many false positives and missing relevant genes. The RNA sequencing data have been deposited in the Gene Expression Omnibus (GEO) database under accession number GSE289632.

### 4.4. Quantitative RT-PCR

Briefly, 1 μg of RNA was reverse-transcribed using random primers to detect selected sncRNAs, following the manufacturer’s instructions (Applied Biosystems, Carlsbad, CA, USA). Quantitative RT-PCR was performed using the SYBR Gene Expression Master Mix (Applied Biosystems). For snoRNA detection, primer design and PCR conditions were employed as previously described [98,99]. The expression levels of sncRNAs were quantified using an Applied Biosystems 7500 Fast Real-Time PCR System, with RNU6B serving as the normalization control. All reactions were conducted in triplicate, and relative expression was calculated using the comparative cycle threshold method (2─ΔΔCT). Results were expressed as fold changes relative to the control group. The primer sequences used for these analyses (5′–3′ direction) are provided in Appendix A.

### 4.5. Statistical Analysis

Throughout the text, results are expressed as mean ± SEM and analyzed using PRISM software GraphPad Prism 10.4.1 (GraphPad, San Diego, CA, USA). The normality of the dataset was assessed using the Kolmogorov–Smirnov test. Since the data in this study did not follow a normal distribution, non-parametric tests were applied for statistical analysis. Comparisons between two groups were conducted using either the Wilcoxon matched-pairs signed-rank test (Figure 4) or the Mann–Whitney test (Figure 5 and Figure 6), as appropriate. Statistical significance was defined as *p* < 0.05.

## Figures and Tables

**Figure 1 ijms-26-01688-f001:**
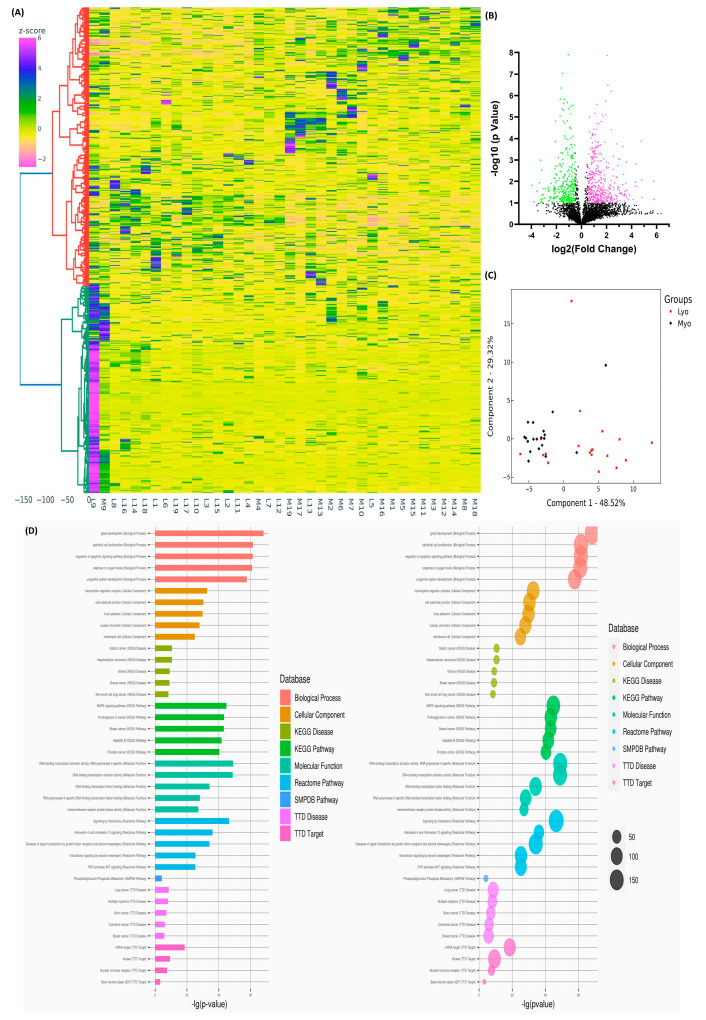
The differential expression of sncRNAs in leiomyoma (Lyo) versus myometrium (Myo). (**A**) Hierarchical clustering heatmap displaying differentially expressed sncRNAs (fold change ≥ 1.5, *p* < 0.05) in 19 paired Lyo and matched myometrial samples. The color gradient represents gene expression levels as z-scores. (**B**) Volcano plot illustrating significantly upregulated (pink; n = 447) and downregulated (green; n = 317) sncRNAs, with a false discovery rate (FDR) *p*-value < 0.05. (**C**) Principal component analysis (PCA) plot of RNA-seq data for paired Lyo and myo samples (n = 19). Each point corresponds to a sample, with myometrial samples (Myo) in black and leiomyoma samples (Lyo) in red. (**D**) Gene Ontology (GO) and KEGG (Kyoto Encyclopedia of Genes and Genomes) pathway enrichment analysis highlighting pathways associated with differentially expressed sncRNAs in Lyo.

**Figure 2 ijms-26-01688-f002:**
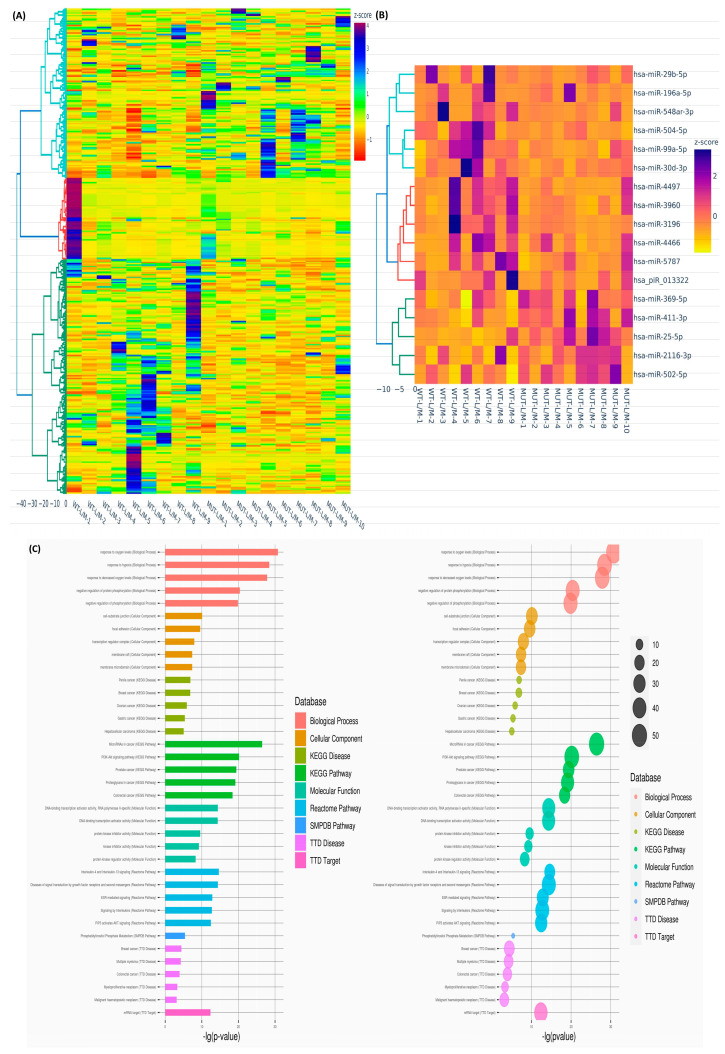
Differential expression of sncRNAs in paired samples stratified by MED12 mutation status. (**A**) Hierarchical clustering heatmap showing the fold change (Lyo/paired Myo) of differentially expressed sncRNAs between MED12-mutated (n = 10) and non-mutated (n = 9) groups (fold change ≥ 1.5, *p* < 0.05). The color gradient represents gene expression levels as z-scores. (**B**) Heatmap depicting 17 enriched genes (Lyo/paired Myo) uniquely identified in the MED12-mutated group (n = 10) but not in the non-mutated group (n = 9) (fold change ≥ 1.5, *p* < 0.05). The color gradient reflects gene expression levels as z-scores. (**C**) Gene Ontology (GO) and Kyoto Encyclopedia of Genes and Genomes (KEGG) pathway enrichment analysis illustrating pathways associated with the 17 differentially expressed sncRNAs.

**Figure 3 ijms-26-01688-f003:**
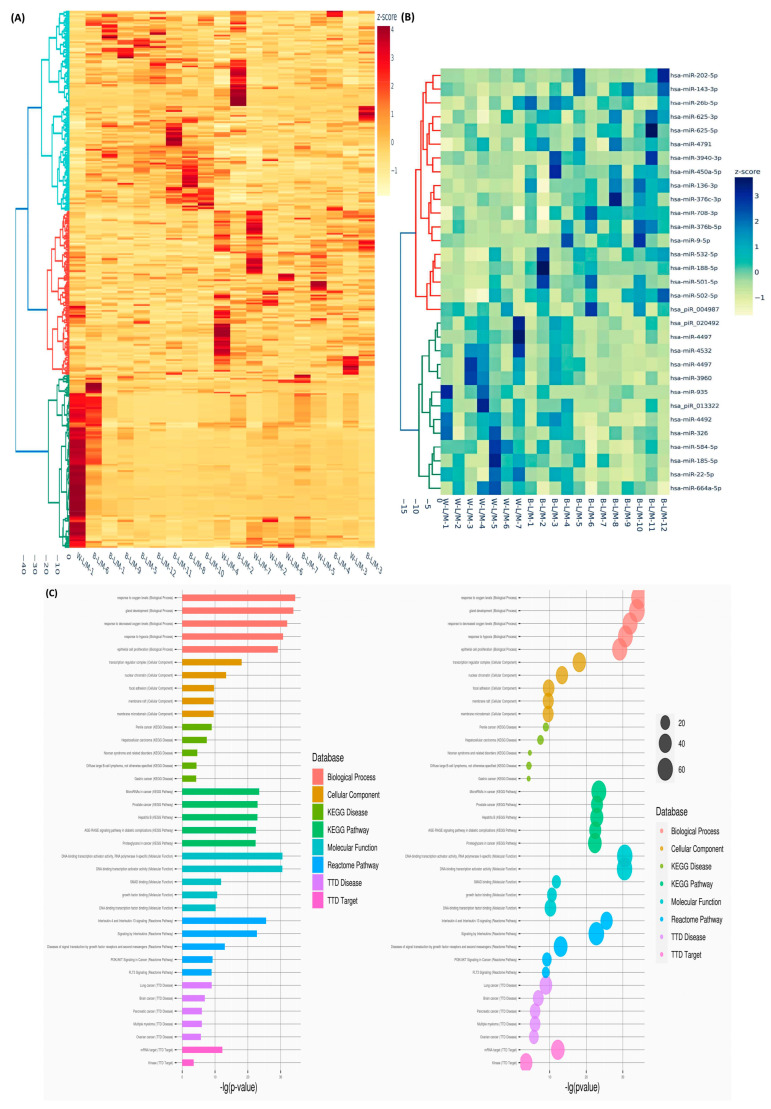
Differential expression of sncRNAs in paired samples stratified by racial groups. (**A**) Hierarchical clustering heatmap displaying the fold change (Lyo/paired Myo) of differentially expressed sncRNAs between Black (n = 12) and White (*n* = 7) groups (fold change ≥ 1.5, *p* < 0.05). The color gradient represents gene expression levels as z-scores. (**B**) Heatmap depicting 31 enriched transcripts (Lyo/paired Myo) uniquely identified in the Black group (n = 12) but not in the White group (n = 7) (fold change ≥ 1.5, *p* < 0.05). The color gradient reflects gene expression levels as z-scores. (**C**) Gene Ontology (GO) and Kyoto Encyclopedia of Genes and Genomes (KEGG) pathway enrichment analysis illustrating pathways associated with the 31 differentially expressed sncRNAs.

**Figure 4 ijms-26-01688-f004:**
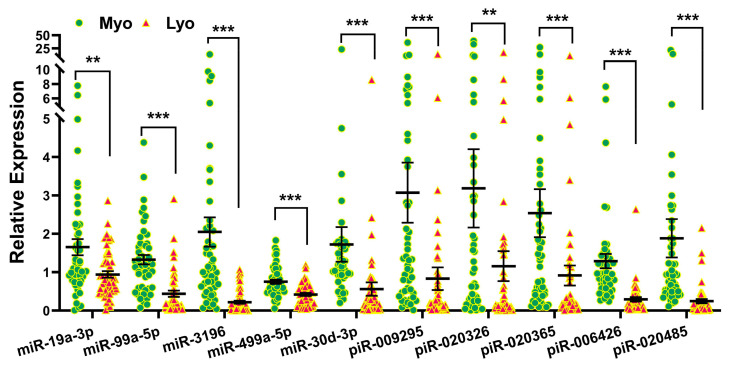
Expression levels of selected sncRNAs in 51 Lyo and paired Myo samples assessed by qRT-PCR. Results are shown as mean ± SEM, with significance levels indicated by asterisks (** *p* < 0.01; *** *p* < 0.001).

**Figure 5 ijms-26-01688-f005:**
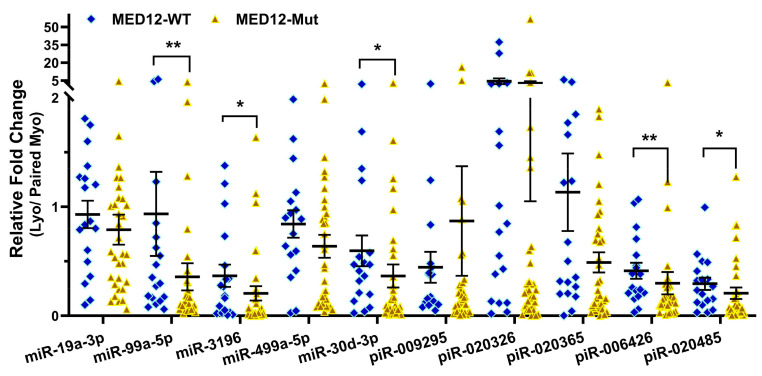
Fold change expression of sncRNAs (Lyo/paired Myo) in MED12-mutated (n = 33) and non-mutated (n = 18) specimens measured by qRT-PCR. Results are presented as mean ± SEM, with significance levels indicated by asterisks (* *p* < 0.05; ** *p* < 0.01).

**Figure 6 ijms-26-01688-f006:**
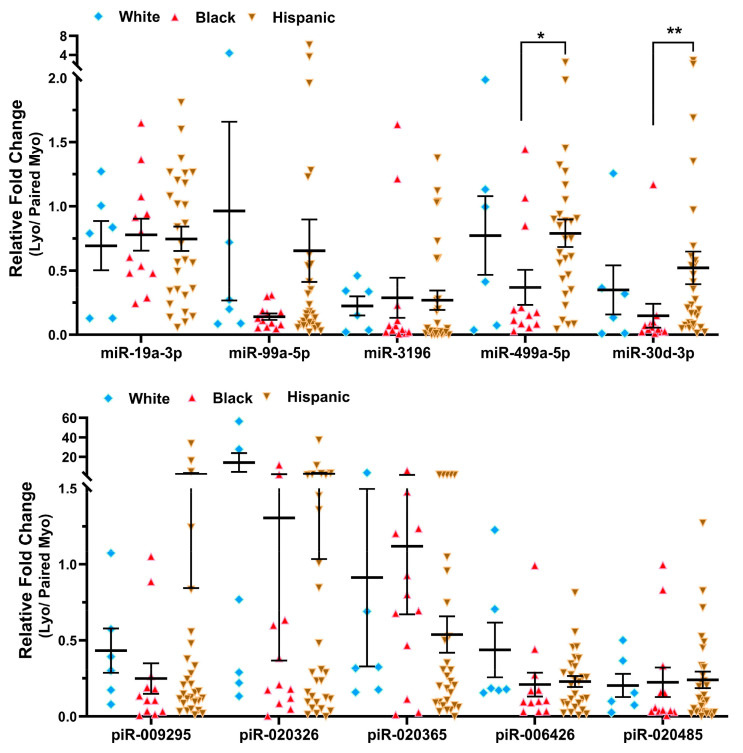
Fold change expression of sncRNAs (Lyo/paired Myo) in White (n = 6), Black (n = 12), and Hispanic (n = 29) groups measured by qRT-PCR. Results are presented as mean ± SEM, with significance levels indicated by asterisks (* *p* < 0.05; ** *p* < 0.01).

## Data Availability

Raw data were generated at The Lundquist Institute. Derived data supporting the findings of this study are available from the corresponding author, O.K., on request.

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
