# Peer review of "Differential Expression of Small Non-Coding RNAs in Uterine Leiomyomas"

_ijms, 2025, doi:10.3390/ijms26041688_

Round 1
Reviewer 1 Report
Comments and Suggestions for Authors
The manuscript “Differential Expression of MED12 Mutation and Race/ethnicity-Associated Small Non-Coding RNAs in Uterine Leiomyomas” investigates the expression profiles of differentially expressed sncRNAs in a large cohort of uterine leiomyomas and matched myometrium samples (stratified by MED12 mutation status and race/ethnicity) using high-throughput sequencing.
The comprehensive data presented herein are valuable and engaging, representing a substantial contribution to the field of research.
The whole manuscript is written understandably, with an appropriate introductory part and a sufficiently explained Materials and Method section. The results are well-explained and presented clearly and understandably, with a proper number of illustrative representations of sufficient quality. The following discussion is adequately written, with a concluding part summarizing the study's significant findings. A corresponding and an up-to-date reference list accompany the manuscript.
Nevertheless, some obstacles must be corrected before the manuscript can be considered for publication.
These include:
Line 85-86…The authors have written: “…MED12 mutation status (mutated: n=10; non-mutated: n=9)…” The same data are presented in the Abstract section. However, in the Materials and Method section, the authors have written: “Among the 19 paired tissues used for next-generation RNA sequencing, 11 leiomyomas were MED12 mutation-positive, and 8 were MED12 mutation-negative.” Please correct or explain.
A minor revision of the manuscript is suggested.
Author Response
The manuscript “Differential Expression of MED12 Mutation and Race/ethnicity-Associated Small Non-Coding RNAs in Uterine Leiomyomas” investigates the expression profiles of differentially expressed sncRNAs in a large cohort of uterine leiomyomas and matched myometrium samples (stratified by MED12 mutation status and race/ethnicity) using high-throughput sequencing.
The comprehensive data presented herein are valuable and engaging, representing a substantial contribution to the field of research.
The whole manuscript is written understandably, with an appropriate introductory part and a sufficiently explained Materials and Method section. The results are well-explained and presented clearly and understandably, with a proper number of illustrative representations of sufficient quality. The following discussion is adequately written, with a concluding part summarizing the study's significant findings. A corresponding and an up-to-date reference list accompany the manuscript.
Nevertheless, some obstacles must be corrected before the manuscript can be considered for publication.
These include:
Line 85-86…The authors have written: “…MED12 mutation status (mutated: n=10; non-mutated: n=9)…” The same data are presented in the Abstract section. However, in the Materials and Method section, the authors have written: “Among the 19 paired tissues used for next-generation RNA sequencing, 11 leiomyomas were MED12 mutation-positive, and 8 were MED12 mutation-negative.” Please correct or explain.
Response: Thanks for the suggestions. Among the 19 paired tissues used for next-generation RNA sequencing, 10 leiomyomas were MED12 mutation-positive, and 9 were MED12 mutation-negative. We have corrected the typo. Thanks.
Reviewer 2 Report
Comments and Suggestions for Authors
In this study, Tsai-Der et al. applied NGS technology to sequence the small non-coding RNA in Leiomyomas of seven while and 12 black men. They have studied the differential expressed genes, the MED12 mutation in race background. They suggest that the abnormal expression of sncRNA may affect key cellular processes such as cell proliferation, and inflammation through dynamic regulatory interactions related to MED12 mutations and ethnic background, thereby promoting the development and progression of Leiomyomas. Before consider publishing in our journal, some revisions must be made.
Comments:
1. The Paragraph in Line 166-210 are too long, please divide to more specific parts.
2. In our previous experience, we always use Fold change≥2 as the cutoff in differentially expressed gene analysis, why the authors use Fold Change≥1.5?
3. In my opinion, tRNA and rRNA have very long nucleotides and cannot be considered as “small” non-coding RNA, please further check or revise the corresponding sentences.
4. There are many types of small non-coding RNA, but the authors only identified the differentially expressed miRNA and piRNA. The authors should discuss why snRNA, snoRNA are not in the results.
5. Please discuss more about the regulation mechanism of identified sncRNAs in Leiomyomas
6. All figures are very unclear. I cannot even see the letters in the figure. Please re-uploaded it
7. Figure 3 were put after Figure 5, please rename the figure.
8. No table in the manuscript. Please added a table describe the specimens
9. Please discuss why the sncRNA have some expression difference in white and black people?
Author Response
In this study, Tsai-Der et al. applied NGS technology to sequence the small non-coding RNA in Leiomyomas of seven while and 12 black men. They have studied the differential expressed genes, the MED12 mutation in race background. They suggest that the abnormal expression of sncRNA may affect key cellular processes such as cell proliferation, and inflammation through dynamic regulatory interactions related to MED12 mutations and ethnic background, thereby promoting the development and progression of Leiomyomas. Before consider publishing in our journal, some revisions must be made.
Comments:
- The Paragraph in Line 166-210 are too long, please divide to more specific parts.
Response: Thanks for the suggestions. We have divided into two parts.
- In our previous experience, we always use Fold change≥2 as the cutoff in differentially expressed gene analysis, why the authors use Fold Change≥1.5?
Response: Thanks for the suggestions. We used 1.5-fold change instead of 2.0. to identify a greater number of differentially expressed genes. We analyzed the data with robust statistical tests including adjusted p-values and FDR corrections. This approach can help balance the trade-off between identifying too many false positives and missing relevant genes.
- In my opinion, tRNA and rRNA have very long nucleotides and cannot be considered as “small” non-coding RNA, please further check or revise the corresponding sentences.
Response: Thanks for the suggestions. Some studies include tRNA and rRNA in the broader category of small RNAs due to their size (tRNA: ~70-90 nucleotides; 5S rRNA: ~ 120 nucleotides) and non-coding. However, recent research classifies tRNA and rRNA as non-coding RNAs but does not typically categorize them as sncRNAs, as sncRNAs primarily refer to regulatory RNAs involved in gene expression modulation rather than translation. tRNA and rRNA are structural and functional RNAs essential for protein synthesis; they do not primarily function as gene regulators. Based on recent findings, we have revised the corresponding sentences accordingly.
- There are many types of small non-coding RNA, but the authors only identified the differentially expressed miRNA and piRNA. The authors should discuss why snRNA, snoRNA are not in the results.
Response: Thanks for the suggestions. Since most differentially expressed sncRNAs identified in this study are miRNAs and piRNAs, we did not select any snRNAs or snoRNAs for validation.
- Please discuss more about the regulation mechanism of identified sncRNAs in Leiomyomas
Response: Thanks for the suggestions. Unfortunately, there isn’t much information of identified sncRNAs in leiomyomas. We have included all associated information in the discussion.
- All figures are very unclear. I cannot even see the letters in the figure. Please re-uploaded it
Response: We have checked and make sure all figures follow the resolution requirement.
- Figure 3 were put after Figure 5, please rename the figure.
Response: We have checked and make sure the figures are in the correct order.
- No table in the manuscript. Please added a table describe the specimens
Response: Thanks for the suggestions. We have included a table in the supplemental file.
- Please discuss why the sncRNA have some expression difference in white and black people?
Response: Thanks for the suggestions. We have added more sentences in the discussion as below:
“Emerging evidence also highlights race-specific dysregulation of sncRNAs in Lyo pathophysiology. Differential expression of miRNAs, including miR-200c, miR-21, and Let-7, has been reported in Lyo from Black women. Additionally, variations in DNA methylation profiles between myometrial samples from Black and White women have been identified, which can influence the expression of miRNAs and other sncRNAs involved in fibroid pathogenesis. Furthermore, given that fibroids are hormone-responsive, racial differences in hormone-related gene expression further exacerbate these disparities. A deeper understanding of these molecular mechanisms is crucial for developing targeted interventions and personalized treatment strategies to address racial disparities in Lyo pathophysiology.”
Round 2
Reviewer 2 Report
Comments and Suggestions for Authors
The quality of the manuscript have been improved a lot, now it can be acceptted for publication.
Author Response
Thank you for your comments.